# Mechanistic Elucidation of the Anti-Ageing Effects of *Dendrobium officinale* via Network Pharmacology and Experimental Validation

**DOI:** 10.3390/foods14193418

**Published:** 2025-10-03

**Authors:** Zhilin Chen, Zhoujie Yang, Shanshan Liang, Weiwei Ze, Zhou Lin, Yuexin Cai, Lixin Yang, Tingting Feng

**Affiliations:** 1College of Pharmacy, Guizhou University of Traditional Chinese Medicine, Guiyang 550025, China; yxykeyanke@163.com (Z.C.); 18684113316@163.com (Z.L.); cyx2131898331@163.com (Y.C.); 2Key Laboratory of Economic Plants and Biotechnology, Kunming Institute of Botany, Chinese Academy of Sciences, Kunming 650201, China

**Keywords:** ageing, antioxidant, compound–target pathway, *Dendrobium officinale*, network pharmacology, traditional Chinese medicine

## Abstract

*Dendrobium officinale* (Orchidaceae) is a commonly used medicinal and edible herb. Although its anti-ageing properties have been demonstrated, the underlying mechanisms remain unclear. We employed network pharmacology and molecular biology techniques to systematically explore its anti-ageing mechanisms. An ageing model was established using D-galactose-induced Kunming mice. *D. officinale* significantly ameliorated ageing-related symptoms, including behavioural impairment and organ index reduction. It enhanced antioxidant capacity by increasing serum T-AOC levels and restoring renal activities of key antioxidant enzymes (SOD, GSH-Px, CAT) while reducing MDA; it suppressed serum TNF-α levels, indicating anti-inflammatory effects. Histopathological examination revealed that *D. officinale* alleviated D-galactose-induced renal damage, including tubular cell swelling and glomerular capsule widening. Network pharmacology identified 8 core active compounds (e.g., 5,7-dihydroxyflavone, naringenin) and 10 key targets (e.g., HSP90AA1, EGFR, MAPK3). KEGG analysis highlighted pathways including neuroactive ligand–receptor interaction, cAMP signalling, and calcium signalling. Molecular docking confirmed strong binding affinities between core compounds and key targets. Western blotting and immunohistochemistry validated that *D. officinale* upregulated EGFR, HSP90AA1, ERK, and GAPDH expression in renal tissues. In summary, *D. officinale* exerts anti-ageing effects by modulating oxidative stress, suppressing inflammation, and regulating multiple signalling pathways. Our findings provide a scientific rationale for its application in anti-ageing interventions.

## 1. Introduction

Ageing, both pathological and physiological, is a complex biological process characterised by a decline in the functions of tissues and organs, structural degeneration, and reduced adaptability and resistance, all of which contribute to the increased morbidity and mortality caused by multiple chronic diseases [1,2]. According to the UN DESA population division, approximately 900 million people worldwide are 60 years or older, and the percentage of such individuals is projected to increase to 21.5% of the global population by 2050 [3]. As ageing progresses, susceptibility to diseases associated with this process, such as vascular ageing disorders [4,5,6], diabetes [7], muscle dysfunction [8,9], macular degeneration [10], Alzheimer’s disease [11,12], skin diseases [13], and numerous other diseases [14,15,16,17,18], increases, posing a severe threat to the physical and mental health of older people.

With ageing, the kidneys undergo complicated functional and structural changes, which can easily lead to pathological changes. Acute kidney injury and chronic kidney disease share many phenotypic similarities with ageing, including cellular senescence, inflammation, fibrosis, sparse blood vessels, glomerular loss, and renal tubular dysfunction [19,20]. Ageing has been suggested as a major cause of increased incidence of chronic renal diseases and acute renal injury [21]. Therefore, decreasing the number of senescent cells is considered a promising therapeutic strategy for reducing ageing-associated renal diseases. Although several theories on the mechanisms of ageing have been proposed, more efforts are needed to explore the exact mechanisms that could guide effective anti-ageing strategies [22,23,24]. Hence, the quest for effective anti-ageing strategies is of paramount importance. Notably, Chinese medicine offers promising anti-ageing remedies because many Chinese herbs contain potent antioxidants and anti-inflammatory compounds that can help combat the oxidative stress and inflammation associated with ageing. Therefore, the exploration of Chinese medicine for the prevention and management of age-related conditions warrants further investigation. Several Chinese herbs contain numerous bioactive metabolites, and their extracts usually have antioxidant, anti-inflammatory, antibacterial, nephroprotective, and other properties [25].

Accumulation of reactive oxygen species (ROS) due to impaired antioxidant defences and mitochondrial dysfunction drives oxidative stress, damaging cellular components (particularly mitochondrial DNA) and promoting apoptosis/necrosis, which accelerates ageing and age-related diseases [26]. D-Galactose (D-Gal, a reducing sugar) is widely used to induce oxidative stress in vivo in order to mimic natural ageing in mice [27,28]. With advancing age, the efficacy of antioxidant defence mechanisms, which involves enzymes, such as superoxide dismutase (SOD), catalase (CAT), and glutathione peroxidase (GSH-Px), in mitigating the biological harm caused by ROS diminishes [29]. Malondialdehyde (MDA), a terminal product of lipid peroxidation in serum and tissue, serves as a sensitive biomarker for assessing the severity of oxidative injury and efficacy of the antioxidant system [30]. Age-related chronic low-grade inflammation (inflammaging) is characterised by elevated circulatory levels of proinflammatory cytokines (e.g., tumour necrosis factor alpha [TNF-α] and IL-6 family members) and chemokines (e.g., MCP-1/CCL2), coupled with increased expression of inflammatory enzymes (e.g., iNOS and COX-2) and endothelial activation markers (e.g., ICAM-1 and VCAM-1) [31,32,33]. Emerging evidence suggests that the equilibrium between proinflammatory mediators and their endogenous antagonists (e.g., IL-1Ra and soluble TNF receptors) critically determines immunosenescence trajectories in ageing populations.

*Dendrobium officinale*, a herb that was first recorded in “Shennong’s Herbal Classic” («神农本草经»J, Dong Han Dynasty, A.D. 25–220), has the ability to strengthen “Yin”, tonify five viscera, nourish the heart, remove arthralgia, relieve fatigue, thicken stomach, lighten the body, prolong life span, and clear heat. According to the literature, this herb contains various bioactive components, such as polysaccharides, bibenzyls, phenanthrenes, and flavonoids, which contribute to a wide range of pharmacological effects, including immune enhancement, antitumor, antioxidant, immunomodulatory, gastrointestinal-protective, and anti-ageing activities [34]. Previous network pharmacology and metabolomic analyses have suggested that certain flavonoids, such as those with high antioxidant capacity, may play a key role in the anti-ageing effects of this herb. For instance, compounds including 5,7-dihydroxyflavone, naringenin, and mangiferin rutin have been proposed as potential active constituents responsible for these benefits [35,36,37]. Although *Dendrobium candidum* is widely used clinically, reports on the effects of its alcohol extract on ageing are limited [38]. Chu et al. [39] found that *D. officinale* exhibits a strong radical scavenging activity in vitro and can ameliorate liver injury in a mouse ageing model via promotion of the Nrf2/HO-1/NQO1 signalling pathway, playing an important anti-ageing role. Liang et al. [40] demonstrated that oral administration of a high dose (1 g/kg) of *D. officinale* juice and a high dose (0.32 g/kg) of *D. officinale* polysaccharide (DOP) for 9 weeks had an anti-ageing effect in D-Gal-induced (0.125 g/kg) ageing in mice, and was associated with significantly increased content of SOD, glutathione peroxidase (GSH-Px), and total antioxidant capacity (T-AOC) in serum, as well as enhanced SOD levels in the heart, liver, kidney, and cerebrum. Similarly, treatment of BALB/c mice with DOP at doses of 50 and 100 mg/kg for 4 weeks exhibited more potent antifatigue activity than did *Rhodiola rosea* extract, which was used as a positive control, as evident from increased triglyceride or fat mobilisation, decreased lipid oxidation, and variability of T and B lymphocytes in the weight-loaded swimming test [41]. *D. officinale* exhibits environment-dependent anti-ageing effects, extending lifespan, enhancing oxidative stress resistance, improving thermal stress tolerance, and mitigating neurodegeneration, likely via antioxidant and stress-protective mechanisms. However, the mechanisms underlying the anti-ageing effects of *D. officinale* remain unclear [42]. Additionally, current pharmacological research lacks component analysis and clinical pharmacological experiments.

Owing to the complexity of the chemical composition and multiple targets of traditional Chinese medicine (TCM), it is difficult to scientifically and comprehensively explain the mechanism underlying their action. Network pharmacology is a method based on the “disease–gene–target–drug” action network, omics data analysis, database retrieval, and virtual computing to systematically explore drug efficacy and the mechanism of action. This methodology has emerged as a robust platform for systematically elucidating the intricate network interactions between bioactive TCM components and their potential therapeutic mechanisms at the systemic level [43].

In this study, we used ultra-high-performance liquid chromatography coupled with quadrupole-Orbitrap high-resolution mass spectrometry (UHPLC-Q-Orbitrap HRMS), combined with multivariate statistical analysis for metabolic profiling of *D. officinale*. Subsequently, network pharmacology approaches were used to identify the potential targets, pathways, and biological functions of *D. officinale* associated with the improved anti-ageing effects. Finally, in vivo experiments were conducted using a D-Gal-induced model of ageing in mice. This study combined the traditional knowledge on the pharmacological effects of *D. officinale* and advanced research technologies to explore the potential mechanisms of action and provide new perspectives for the development of drugs and functional foods as well as for clinical applications. The overall research methodology used to investigate the anti-ageing effects of *D. officinale* is illustrated in Figure 1.

## 2. Materials and Methods

### 2.1. Chemicals and Materials

The CAT assay kit was purchased from Solarbio Biotechnology Co., Ltd. (Beijing, China). Trypsin and kits for GSH-Px activity, SOD assay, T-AOC assay, and MDA assay were obtained from the Nanjing Jiancheng Bioengineering Institute (Nanjing, China). The TNF-α ELISA kit was acquired from Wuhan Huamei Biotechnology Co., Ltd. Antibodies against heat-shock protein (HSP), extracellular regulated protein kinase (ERK), and glyceraldehyde-3-phosphate dehydrogenase (GAPDH) were purchased from Wuhan Huamei Biotechnology Co., Ltd (Wuhan, China), whereas the epidermal growth factor receptor (EGFR) antibody was purchased from Bioss Antibodies Co., Ltd (Beijing, China). The bicinchoninic acid (BCA) protein assay kit and radioimmunoprecipitation assay lysis buffer were procured from GBCBIO Technologies Co., Ltd (Guangdong, China), and Dalian Meilun Biotechnology Co., Ltd (Dalian, China), respectively. D-Gal was procured from Inarco S.p.A. (Milan, Italy). All the other chemicals and reagents were of analytical grade.

The plant name *D. officinale* was checked with “World Flora Online” (www.worldfloraonline.org) or MPNS (http://mpns.kew.org). DOG (*D*. *officinale* stems dried at 70 °C) and Tiepi Fengdou (curled dried *Dendrobium*) were procured from Lingya Original Dendrobium officinale Technology Co., Ltd. (Guangnan County, Yunnan Province, China). The DOP (dried strips processed by stir-frying at 150 °C for 10 min, shaping at 60 °C, and drying at 90 °C) were prepared from the same batch of fresh stems. All materials were authenticated as fresh stems of *D. officinale* Kimura et Migo (Orchidaceae) by Senior Engineer Yang Lixin at the Kunming Institute of Botany, Chinese Academy of Sciences.

### 2.2. Network Pharmacology Analysis

#### 2.2.1. Collection and Screening of Chemical Components in *D. officinale*

Our previous metabolomic study revealed that TPFD contains 370 identifiable metabolites [41]. We systematically analysed these constituents using a network pharmacology approach. After excluding compounds without clearly defined targets based on literature verification, 127 bioactive ingredients were shortlisted (see Appendix A for the complete list).

#### 2.2.2. Screening of Ageing-Related Targets

To systematically identify ageing-related molecular targets, we investigated three established disease-gene databases: GeneCards (https://www.genecards.org/, version 5.12), Comparative Toxicogenomics Database (http://ctdbase.org/, updated 2023), and Online Mendelian Inheritance in Man (https://omim.org/). Using “aging” as the primary search term, we retrieved all associated targets (*n* = 27,783), followed by removal of duplicate entries (*n* = 1085 remaining). The protein targets of the *D. officinale* bioactive compounds (*n* = 248) were cross-referenced with ageing-related targets through Venn analysis using Venny 2.1 (https://bioinfogp.cnb.csic.es/tools/venny/ (accessed on 7 May 2025)), which yielded 248 putative anti-ageing targets (Figure 2). All gene symbols were standardised using UniProtKB (https://www.uniprot.org/, version release 2024_01) to ensure consistency in nomenclature.

#### 2.2.3. Protein–Protein Interaction Network Construction

To construct a functional enrichment model of the protein–protein interaction (PPI) network, the obtained intersecting target genes were imported into the STRING database (https://string-db.org/), “Multiple protein” was selected, the species was set as “*Homo sapiens*”, and the minimum interaction threshold was set at high confidence (0.700). The results were exported to the Cytoscape software (version 3.10.0) for visual analysis. Targets that passed the degree value (Degree) were screened as the core targets. The main active ingredients of *D. officinale* were obtained, target genes for the anti-ageing effects of *D. officinale* and the pathways involved were matched, and the “drug–component–target–pathway” network diagram was constructed by using the Cytoscape software. The core components were screened based on three indicators: degree, betweenness centrality (BC), and closeness centrality (CC).

#### 2.2.4. Gene Ontology and Kyoto Encyclopedia of Genes and Genomes Analyses

Using the Metascape database (https://metascape.org/gp/index.html#/main/step1 (accessed on 7 May 2025)), the target genes of *D. officinale* anti-ageing (i.e., intersection genes) were subjected to Gene Ontology (GO) annotation and Encyclopedia of Genes and Genomes (KEGG) pathway enrichment analyses. In the GO analysis, the top 20 results for biological process (BP), molecular function (MF), and cellular composition (CC) were selected based on *p* < 0.05 using the We Bio website (http://www.bioinformatics.com.cn/) and a histogram was generated. In the KEGG analysis, the criterion of *p* < 0.05 was also used for screening, and the first 20 representative pathways were selected to draw the KEGG bubble map.

#### 2.2.5. Molecular Docking Methodology

Molecular docking was performed using the Auto Dock software (AutoDock 4.2.6). First, the 3D structure of the core components was obtained from the PubChem database and converted into the corresponding MOL2 format using the Open Babel software (Open Babel 3.1.1). Thereafter, the PDB ID and protein structure of the core target were found and downloaded from the PDB protein database (https://www.rcsb.org/). The pdb file of the core components was imported into the PyMOL software (PyMOL 2.5.2) to remove water molecules and ligands, and the ligand molecules used in subsequent docking were obtained. Using the same software, water molecules, ligands, and other small molecule structures were removed from the 3D protein structure of the core target and the receptor protein was further processed using Auto Dock Tools for hydrogenation and charge. The ligand and receptor proteins were verified using the Auto Dock software, and the corresponding binding energies were obtained. Finally, the PyMOL software was used to visually analyse the docking mode.

### 2.3. Experimental Verification

#### 2.3.1. Samples and Sample Preparation

DOG, Tiepi Fengdou, and DOP coarse powders (1.5 kg of each) were subjected to sequential ethanol reflux extraction under standardised conditions: (1) initial extraction with 8× volume of 79% ethanol (24 h soaking followed by 48 min reflux), (2) secondary extraction with 6× volume of 79% ethanol (48 min reflux), and (3) final extraction with 4× volume of 79% ethanol (48 min reflux). The combined filtrates were concentrated under reduced pressure at 70 °C using rotary evaporation, with the resultant extract stored for subsequent aqueous reconstitution at required concentrations. Control solutions were prepared by dissolving (a) 3 g vitamin E in 300 mL distilled water and (b) 24 g D-Gal in 300 mL normal saline (800 mg/mL), both of which were stored at 4 °C until use.

#### 2.3.2. Animal Experiments

Specific-pathogen free-grade male Kunming mice (6 weeks old, 20 ± 2 g body weight) were procured from Changsha Tianqin Biotechnology Co., Ltd (located in Changsha, China). (Animal License No. SCXK(Xiang)2022-0011). All animal experiments were conducted in accordance with the ARRIVE guidelines 2.0 (https://arriveguidelines.org) and complied with the UK Animals (Scientific Procedures) Act 1986 and associated guidelines, European Union Directive 2010/63/EU for animal experiments, and the National Research Council’s Guide for the Care and Use of Laboratory Animals (8th edition). Protocols were approved by the Institutional Animal Care and Use Committee of Guizhou University of Traditional Chinese Medicine (approval no. GZU-20220038, approved on 20 June 2022). All efforts were made to minimise animal suffering and reduce the number of animals used.

The animals received standard chow and water ad libitum throughout the study. Following the standards for conducting pharmacological experiments, the mice were randomly allocated into six groups: (1) normal control (n = 10, i.p. administration of 0.9% saline); (2) model group (daily i.p. injection of 800 mg/kg D-Gal); (3) DOG, Tiepi Fengdou, and DOP sample groups (oral gavage at 1.56 or 6.24 g/kg/d, combined with 800 mg/kg D-Gal i.p. injection); and (4) vitamin E group (VE, 10 mg/kg/d oral administration). The basis for determining the experimental doses of the extract was derived from commonly used clinical doses and converted via body surface area coefficient-based equivalence scaling. All intraperitoneal injections and oral gavages (0.1 mL/10 g dosing volume) were administered at fixed timepoints daily for 30 consecutive days, with weekly body weight monitoring during the intervention period.

The doses of *D. officinale* used in this study (6.24 and 1.56 g/kg/d for the high- and low-dose groups, respectively) were determined based on the conversion of clinically relevant doses via body surface area coefficient-based equivalence scaling. The common clinical daily dose for adults (with a standard body weight of 60 kg) ranges from 3 to 12 g/person/day [44,45]. According to FDA guidelines, the human clinical dose was converted to the mouse equivalent dose using a body surface area normalisation factor of 12.3. Consequently, the low dose (1.56 g/kg/d) corresponds to a human dose of 3 g/person/day (50 mg/kg/d), whereas the high dose (6.24 g/kg/d) corresponds to a human dose of 12 g/person/day (200 mg/kg/d). This dose range was selected to cover and slightly exceed the clinical equivalent to adequately evaluate the dose–effect relationship and pharmacological activity of DOG in the animal model.

#### 2.3.3. Measurement of Physiological Indicators

##### Body Weight Measurement

During the modelling period, the growth status of mice, including diet, faeces, activity, and weight changes, was recorded. The average body weight of the mice was calculated using the following formula: average weekly body weight of mice = total weekly body weight of mice in the group/number of mice. A trend chart of body weight changes was created.

##### Organ Index Measurement

After euthanasia via cervical dislocation, the mice were dissected and the heart, liver, spleen, lungs, kidneys, and brain tissues were collected. The organs were cleaned to remove any residual blood, and aluminium foil was used to absorb any remaining moisture. The organs were weighed, and their weights were recorded. The organ index was calculated using the following formula: organ index (%) = organ wet weight/mouse body weight × 100.

### 2.4. Determination of Biochemical Parameters

Following euthanasia, renal tissues were homogenised in ice-cold phosphate-buffered saline (1:9, w/v) and centrifuged (3000× *g*, 10 min, 4 °C). The supernatant was divided for (i) analysis of oxidative stress parameters (SOD at 450 nm, MDA at 532 nm, GSH-Px at 412 nm, and CAT, which were estimated by measuring the absorbance at 450, 532, 412, and 520 nm, respectively) using standardised commercial kits and (ii) protein quantification via BCA assay for data normalisation. For systemic evaluation, terminal blood samples collected via retro-orbital puncture under anaesthesia were processed (3500 × *g*, 10 min) to obtain serum for TNF-α, and T-AOC was measured using the total antioxidant capacity assay kit. All assays were performed in triplicate using a microplate reader with the appropriate wavelength settings.

### 2.5. Western Blotting Analysis

The protein expression levels of HSP90, EGFR, ERK, and GAPDH were determined using Western blotting. Briefly, the kidney tissue was lysed using a tissue lysis/extraction reagent in the presence of a 1× protease inhibitor cocktail. Protein concentrations were determined using a BCA kit. The prepared protein samples and marker were added to the loading wells with a microsampler; the total protein content in each sample was 40 μg. The proteins were then transferred onto polyvinylidene difluoride membranes. After blocking, the membranes were incubated overnight at 4 °C with primary antibodies specific for HSP90 (1:1000), EGFR (1:1000), ERK (1:1000), and GAPDH (1:1000). The membranes were washed and incubated for 2 h at room temperature (22–25 °C) with horseradish peroxidase-conjugated secondary antibodies. The protein bands were developed using the ECL and imaged using an Imaging System, and the grey value of the film was analysed using Image-Pro Plus 4.5 for quantitation.

### 2.6. Histologic Examination

The renal tissues were fixed in 4% formaldehyde, embedded in paraffin, and cut into 5 µm sections. The sections were stained with haematoxylin and eosin. Changes in pulmonary histopathology were visualised under a microscope.

### 2.7. Basic Methodology Description

Renal tissue sections were processed for immunohistochemical analysis to evaluate the localisation and expression of EGFR, HSP90, ERK, and GAPDH in *D. officinale*-treated, D-Gal-induced ageing mice. Following antigen retrieval (using EDTA buffer, pH 9.0) and blocking, the sections were incubated with primary antibodies (4 °C, overnight) and horseradish peroxidase-conjugated secondary antibodies, and then developed with 3,3′-diaminobenzidine. Mayer’s haematoxylin counterstaining was performed, and protein expression (brown) was quantified based on mean optical density using IPP6.0.

### 2.8. Statistical Analysis

All tests and analyses were repeated at least three times. The experimental results are presented as the mean ± standard deviation (mean ± SD). *p*-Values were calculated via the Student’s *t*-test or one-way analysis of variance (ANOVA). Statistical analyses were performed using IBM SPSS Statistics version 26.0. The significance of each group was verified using one-way ANOVA. *p*-Value < 0.05 was considered significant, whereas *p*-value <0.01 represented extreme statistical significance.

## 3. Results

### 3.1. Active Compounds and Target Screening

Using metabolomic analysis, we identified 370 metabolites in *D. officinale,* including lipids, phenylpropanoids, polyketides, and nitrogen-containing organo-oxygen compounds (Appendix A). The 2D structures of the chemical components of *D. officinale* were imported into the online Swiss Target Prediction database to predict the targets, with a probability > 0.13 criterion. A total of 265 potential target molecules with canonical SMILES structures were ultimately identified (Appendix A). Moreover, 1085 common anti-ageing target proteins were identified after removing overlapping targets among different databases, with 62 targets from the Comparative Toxicogenomics Database, 601 targets from the Online Mendelian Inheritance in Man database, and 27120 targets from the GeneCards database. We then crossed 265 *D. officinale* active component targets with 1085 anti-ageing-related targets using the VENNY 2.1 software (Figure 2). Ultimately, 248 genes were identified as targets for both the active ingredients and ageing-related genes (Appendix A).

### 3.2. PPI Network Analysis and Key Target Identification

The 248 common targets were analysed using STRING version 11.0 to construct a PPI network (Figure 3), which comprised 216 nodes and 750 edges with an average degree value of 6.94, where nodes represent target proteins and edges represent interactions between proteins. In this network interaction, the larger the degree, the stronger the interaction between the targets. This also indicates that the particular key target protein plays a pivotal role in regulating the network as a hub target. Then, the Cytoscape 3.7.2 software was used for the visualisation and calculation of topological parameters, such as degree, BC, and CC. Moreover, the top 10 targets were obtained using the degree value as the condition for screening core targets, mainly involving HSP90AA1, SRCE, GFR, MAPK3, EP300, GAPDH, ESR1, APP, RELA, and PPARG. The core targets are listed in Appendix A.

### 3.3. GO and KEGG Analyses

The Metascape database was used to perform GO enrichment analyses for biological BP, CC, and MF, as well as KEGG pathway analysis of a set of 248 common target genes. In the BP category, the most significantly enriched genes were associated with steroid hormone-mediated signalling pathways, intracellular receptor signalling pathways, and cellular responses to dopamine and catecholamine stimulation. Within the CC group, the most enriched genes were localised in glutamatergic synapses, interneuronal synapses, postsynaptic density, and other pre- and postsynaptic regions. The genes most prominently enriched in the MF category were also related to steroid hormone-mediated signalling pathways, intracellular receptor signalling pathways, and cellular responses to dopamine and catecholamine stimulation (Figure 4). KEGG pathway analysis revealed that the pathways involved in neuroactive ligand–receptor interactions, cancer, senile dementia, chemical carcinogenesis-receptor activation, calcium signalling, cAMP signalling, and the renin–angiotensin system are implicated in the regulation of these common target genes (Figure 5).

### 3.4. Identification of Core Active Ingredients

Based on the predicted results of the identified active components, key targets, and the top 20 related pathways using the KEGG database, an integrated component–target–pathway network was constructed using Cytoscape (Figure 6). The figure shows 310 nodes and 1138 edges, with an average degree value of 7.342, of which the lilac oval represents 107 active ingredients, the small orange diamond represents 181 targets, and the blue square represents 20 pathways.

By sorting the results based on degree, BC, and CC, we identified eight core active components: 5,7-dihydroxyflavone, mangiferic acid, norartocarpanone, naringenin, sideritoflavone, 4′,5,8-trihydroxyflavanone, benzyl gentiobioside, and rutin (see Appendix A for details). Among these, 5,7-dihydroxyflavone exhibited the highest significance (degree = 45; BC = 0.071; CC = 0.429), followed by mangiferic acid (degree = 15; BC = 0.027; CC = 0.390). Additionally, the top 10 key targets in the network included HSP90AA1, SRC, EGFR, MAPK3, EP300, GAPDH, ESR1, PP, RELA, and PPARG.

### 3.5. Molecular Docking

Molecular docking was conducted using AutoDock 4.0 to investigate the interactions between the key active components of *D. officinale* (specifically, 5,7-dihydroxyflavone, mangiferic acid, norartocarpanone, naringenin, sideritoflavone, 4′,5,8-trihydroxyflavanone, benzyl gentiobioside, and rutin) and key target proteins (HSP90AA1, EGFR, MAPK3, EP300, GAPDH, ESR1, APP, and RELA). The interpretation of binding energies was as follows: binding energy < 0 kJ/mol indicates a spontaneous interaction, binding energy ≤ −5 kJ/mol (equivalent to −1.2 kcal/mol) suggest a favourable binding effect, and binding energies ≤ −7 kJ/mol (equivalent to −1.68 kcal/mol) denote strong binding activity. A binding energy threshold of ≤ −1.68 kcal/mol was used as the screening criterion, based on a previously established protocol [42,43,44] for identifying potential bioactive compounds from natural products in silico. The results are summarised in Appendix A.

We observed that 5,7-dihydroxyflavone exhibited robust binding affinity toward all 10 core targets, forming a stable conformation with low energy. Notably, 5,7-dihydroxyflavone demonstrated exceptional docking and affinity for HSP90AA1 and EGFR, which may suggest a potential therapeutic role of *D. officinale* in anti-ageing applications. Among the targets, 5,7-dihydroxyflavone exhibited the lowest docking energy with HSP90AA1 (−6.72 kJ/mol), indicating superior binding ability and stability compared with those of the other targets. In contrast, mangiferic acid exhibited relatively high binding energy with HSP90AA1, indicative of weaker binding affinity. Naringenin and 4′,5,8-trihydroxyflavanone also demonstrated strong binding activity to multiple targets, including HSP90AA1, EGFR, MAPK3, EP300, GAPDH, ESR1, APP, and RELA. The binding efficacy of some flavonoids to their targets was somewhat inferior. In particular, benzyl gentiobioside and rutin generally failed to bind effectively to the core targets.

The docking models of the top six molecules with superior binding energies are illustrated in Figure 7, providing visual insights into their interactions with the target proteins. These findings offer preliminary insights into the potential molecular mechanisms by which *D. officinale* and its active constituents may exert therapeutic effects, although these proposed interactions remain predictive and require further experimental validation.

### 3.6. Effect of D. officinale on Body Weight and Organ Index

We explored the anti-ageing effects of *D. officinale* alcohol extract (DOAE) prepared from dried DOG, TPFD, and DOP in 70% alcohol. Mice treated with D-Gal for 7 weeks displayed significant signs of ageing, such as decreased vitality, lethargy, and hair loss, compared with the control group. However, treatment with DOG, TPFD, or DOP effectively reversed these age-related changes in mice. As shown in Figure 8, the model group treated with D-Gal experienced a significant reduction in body weight compared with that of the control group (*p* < 0.01). In contrast, mice in the VE group exhibited a significant increase in weight gain compared with those in the D-Gal group (*p* < 0.01). Moreover, the organ indices for the heart, liver, spleen, kidney, lung, and brain in the D-Gal-treated model group were significantly lower than those in the control group (*p* < 0.01). Notably, high-dose treatments (6.24 g/kg/d) resulted in significantly greater recovery in organ indices compared with those in low-dose (1.56 g/kg/d) D-Gal-induced ageing mice (*p* < 0.01). However, supplementation with DOG, TPFD, and DOP significantly alleviated the changes induced by D-Gal treatment compared with those in the model group (*p* < 0.05). Although no significant differences in the organ indices were noted among the DOG, TPFD, and DOP groups (Table 1), these results collectively highlight the significant anti-ageing effects of DOG, TPFD, and DOP in this ageing model.

### 3.7. Alcoholic Extracts of D. officinale Alleviate D-Galactose-Induced Oxidative Stress and Inflammation by Modulating Antioxidant Enzyme Activities and Suppressing TNF-α Levels

To assess the antioxidant and anti-inflammatory properties of *D. officinale*, we measured serum and renal tissue biomarkers in D-Gal-induced ageing rats. In particular, we evaluated serum T-AOC (T-AOC was expressed as Trolox equivalents) and TNF-α levels, along with renal tissue concentrations of MDA, GSH-Px, CAT, and SOD. Exposure to D-Gal significantly elevated the production of proinflammatory cytokines, such as TNF-α, while simultaneously reducing systemic T-AOC (*p* < 0.05). Notably, both alcoholic extracts of *D. officinale* and vitamin E, which served as a positive control, effectively suppressed the release of TNF-α (*p* < 0.05; Table 2). Additionally, the oxidative stress induced by D-Gal was characterised by increased MDA levels and decreased SOD, CAT, and GSH-Px activities (*p* < 0.05; Table 3). High-dose extracts demonstrated superior efficacy in restoring antioxidant enzyme activities (SOD, CAT, GSH-Px) and reducing TNF-α levels compared with that of low-dose groups (*p* < 0.01). Treatment with *D. officinale* extract or VE significantly restored the activity of these antioxidant enzymes (*p* < 0.05), indicating their potential protective effects against oxidative damage.

### 3.8. Effects of Different D. officinale Extracts on the Expression of Key Proteins in D-Galactose-Induced Ageing Kidney

Based on four key targets (EGFR, HSP90AA1, ERK, and GAPDH) screened through network pharmacology analysis, we employed Western blotting to examine the effects of *D. officinale* and aqueous VE on the expression of these proteins in the kidney tissue of D-Gal-induced ageing mice models. Compared with those in the normal control group, the protein levels of EGFR, HSP90AA1, ERK, and GAPDH were significantly downregulated in the D-Gal model group (*p* < 0.01; Figure 9 and Figure 10). After intervention with different doses of the alcoholic extract of *D. officinale*, all treatment groups showed dose-dependent upregulation of these proteins. A significant improvement was observed in the low-dose group (*p* < 0.05), with more pronounced effects observed in the high-dose group (*p* < 0.01). Western blot analyses confirmed dose-dependent upregulation of EGFR, HSP90AA1, ERK, and GAPDH in kidney tissues, with high-dose groups exhibiting effects comparable to those of the vitamin E positive control (*p* < 0.01). Notably, the VE group exhibited the most remarkable intervention effect, exhibiting significantly better recovery of protein expression than that of the other extract groups (*p* < 0.01), followed by the high-dose TPFD group.

### 3.9. Effects of Different D. officinale Extracts on Alleviating Morphological Damage to the Kidneys of D-Gal-Induced Ageing Rats

Haematoxylin and eosin staining was performed to assess histopathological alterations following chronic intraperitoneal administration of D-Gal and subsequent treatment. As shown in Figure 11A, the renal tissue of the normal control group exhibited uniform staining, with renal tubules arranged in an orderly fashion, normal lumina, and a ring-shaped cellular distribution. The renal pelvis cavity was clearly defined and well organised, whereas the glomeruli appeared round, with distinct edges, regular cellular arrangements, and darker, clearly visible nuclei. No pathological changes were observed in this patient group. In contrast, the model group displayed notable swelling of tubular cells and glomeruli. Tubular cells exhibited granular and vacuolar degeneration, and the glomerular capsule widened (Figure 11B). These findings indicate that renal tissue sustained damage from D-Gal, confirming the successful establishment of an ageing model. High-dose treatments (particularly TPFD) markedly alleviated renal morphological damage, showing near-normal tissue architecture, whereas low-dose groups exhibited partial improvement.

Following oral administration of *D. officinale*, significant improvements in the aforementioned damages were observed in the low-dose (1.56 g/kg/d) group; however, swollen tubular cells and widened glomerular capsules were still evident. In the high-dose (6.24 g/kg/d) group, swollen tubular cells were absent, and glomeruli with narrowed capsular spaces were observed. Histopathological findings in the kidney tissue in the VE and high-dose TPFD groups (Figure 11E,I) was comparable to those in the control group. These results indicate that *D. officinale* can effectively ameliorate D-Gal-induced renal lesions. Furthermore, D-Gal-induced histopathological lesions showed a dose-dependent response to *D. officinale* treatment.

### 3.10. Immunohistochemical Analysis of the Renal Tissue

To comprehensively investigate the effect of *D. officinale* on protein expression in the renal tissue in mice, we conducted immunohistochemical analyses to examine the localisation and intracellular abundance of EGFR, HSP90, ERK, and GAPDH. All four proteins were expressed in the renal tissue cells and were predominantly localised in the cytoplasm and cell membrane, with only negligible amounts present in the nucleus. In the model group, the mean optical density (MOD) values for EGFR, HSP90, ERK, and GAPDH proteins in the renal tissue were significantly lower than those in the normal control group (*p* < 0.01). Immunohistochemical analyses confirmed dose-dependent upregulation of EGFR, HSP90AA1, ERK, and GAPDH in kidney tissues, with high-dose groups exhibiting effects comparable to the vitamin E positive control (*p* < 0.01). Both the VE and high-dose *D. officinale* groups exhibited significantly elevated MOD values (*p* < 0.01), indicating the positive effect of VE and high-dose *D. officinale* on protein expression. Furthermore, a dose-dependent relationship was observed in MOD values of target proteins between the low- and high-dose *D. officinale* groups. The results of immunohistochemical analysis and statistical comparisons are shown in Figure 12 and Figure 13, respectively. These findings provide valuable insights into the effect of *D. officinale* on protein expression in mouse renal tissue and highlight the potential therapeutic benefits of VE and high-dose *D. officinale* in this context.

## 4. Discussion

Ageing and its associated pathologies represent a considerable global health burden that substantially compromises the quality of life of older people. Consequently, elucidating the molecular mechanisms underlying ageing and developing effective interventions have become crucial research priorities. Investigations into renal ageing not only benefit geriatric health but also provide critical insights for preventing and managing age-related kidney disorders [46,47,48,49]. Traditional Chinese herbal medicines, particularly *D. officinale*, have demonstrated considerable potential in mitigating ageing-associated pathologies [50]. As documented in the classical “Shennong Materia Medica”, this medicinal herb possesses sweet and neutral properties, with traditional applications for “treating internal injuries, resolving qi stagnation, replenishing deficiencies, and strengthening yin”; notably, long-term administration is reported to “fortify gastric function, enhance physical vitality, and promote longevity”. Contemporary pharmacological research has validated these traditional claims, revealing multifaceted bioactivities of *D. officinale*, including potent antioxidant, antiapoptotic, hypoglycaemic, immunomodulatory, antitumor, and anti-ageing effects [51].

Network pharmacology has emerged as a powerful integrative approach that combines the principles of systems biology, network analysis, bioinformatics, and polypharmacology. This methodology aligns exceptionally well with the holistic paradigm of TCM [52]. In recent years, D-Gal-induced ageing models have been widely used to evaluate the anti-ageing efficacy of natural compounds [53]. Substantial evidence indicates that D-Gal accelerates the ageing process in animal models through multiple mechanisms, including the induction of oxidative stress, promotion of neuroinflammation, and triggering of irreversible apoptotic pathways, all of which collectively contribute to accelerated senescence [54,55]. Extensive research has established the pivotal role of ROS-mediated oxidative damage in the ageing trajectory of higher organisms. Particularly relevant are observations from various age-associated disease models that ROS and oxidative stress not only modulate the onset of age-related pathologies but may also fundamentally influence both the rate of ageing and organismal lifespan. Supporting this notion, Li et al. [56] reported that both *Agaricus bisporus* and its purified fractions exhibit significant anti-ageing effects in D-Gal-treated animals, potentially through mechanisms involving the upregulation of antioxidant enzymes, reduction of lipid peroxidation, improvement of organ function, and modulation of lipid metabolism. Similarly, Liang et al. [57] demonstrated pronounced anti-ageing properties of *D. officinale* using D-Gal models, attributing the effects to the regulation of antioxidant defences and immune system modulation. Complementary findings by other researchers show that resveratrol can substantially enhance the viability of aged cells and ameliorate pathological conditions in ageing mice, primarily through the suppression of MDA generation while simultaneously boosting the activities of SOD and CAT [58].

In this study, we systematically identified eight principal bioactive constituents of *D. officinale* (5,7-dihydroxyflavone, mangiferic acid, norartocarpanone, naringenin, sideritoflavone, 4′,5,8-trihydroxyflavanone, benzyl gentiobioside, and rutin) that interacted with crucial molecular targets (HSP90AA1, EGFR, MAPK3, EP300, GAPDH, ESR1, APP, and RELA) to mediate their anti-ageing effects. Mechanistically, these components modulate specific targets, including HSP90AA1, SRC, EGFR, and MAPK3, thereby regulating diverse biological pathways, including neuroactive ligand–receptor interactions, oncogenic signalling cascades, Alzheimer’s disease-related processes, calcium-mediated signalling, and serotonergic synaptic transmission. Of particular interest, the identified active compounds chrysin [59,60], naringenin [61], and 4′,5,8-trihydroxyflavonoids [41] possess well-characterised anti-inflammatory properties.

EGFR, a prominent member of the HER family, plays an essential role in dermal fibroblast dynamics and mitotic regulation. As a membrane-associated receptor expressed on epithelial cells, EGFR is strongly associated with cutaneous ageing processes [62]. Notably, whereas EGFR signalling attenuation promotes ageing phenotypes in glandular and epithelial tissues, its activation stimulates cellular proliferation and suppresses senescence [63]. Similarly, the molecular chaperone HSP90AA1 participates in immune regulation, apoptotic control, and various tumour-related processes [64]. The multifunctional enzyme GAPDH exhibits context-dependent behaviour, with elevated expression associated with proliferative and tumorigenic processes, oxidative conditions, cellular senescence, and apoptotic pathways [65]. Our experimental analyses of renal tissue from D-Gal-induced ageing mice revealed that *D. officinale* extract treatment significantly upregulated the expression of EGFR, HSP90AA1, ERK, and GAPDH proteins (Figure 9), which was in excellent agreement with our network pharmacology predictions.

Comprehensive KEGG pathway analysis established robust connections between the identified targets and two major pathway categories: oxidative stress responses and inflammatory signalling cascades. In particular, these included neuroactive ligand–receptor interactions, oncogenic pathways, Alzheimer’s disease-related mechanisms, and calcium-mediated signalling networks, collectively highlighting the central involvement of inflammatory processes in ageing. During senescence, neurotransmitter receptor populations undergo quantitative changes that are frequently correlated with functional alterations [66]. Mounting evidence suggests that disturbances in calcium homeostasis, particularly mitochondrial calcium overload and electron transport chain complex modifications, markedly increase cellular susceptibility to oxidative damage during ageing. Age-related impairment in calcium signalling contributes to mitochondrial dysfunction, which is characterised by respiratory uncoupling and excessive ROS generation. Lipofuscin accumulation in senescent cells compromises the macroautophagic clearance of damaged mitochondria, establishing a self-perpetuating cycle of oxidative damage, genomic instability, and ultimately programmed cell death [67]. The transcription factor NF-κB orchestrates expression of numerous inflammatory mediators, including TNF-α, which features prominently in D-Gal-induced ageing phenotypes [68]. Notably, ROS exhibit biphasic modulation of NF-κB signalling, with either activating or inhibitory effects depending on temporal and contextual factors. Correspondingly, the NF-κB pathway itself demonstrates dual functionality under oxidative conditions and is capable of exerting both antioxidant and pro-oxidant effects [69]. Collectively, our findings establish that the anti-ageing properties of *D. officinale* emerge from complex multicomponent, multitarget, and multi-pathway interactions.

In our experimental models, DOAE consistently demonstrated significant biological activity, which was consistent with its putative anti-ageing effects. The ageing process characteristically involves elevated levels of proinflammatory mediators, with TNF-α representing a particularly crucial regulator of pulmonary inflammatory responses in senescence [70]. DOAE pretreatment effectively attenuated cytokine (specifically TNF-α) concentrations, indicating substantial anti-inflammatory efficacy in D-Gal-induced ageing rats. Given the well-established role of oxidative stress in ageing pathophysiology, we further demonstrated that DOAE treatment robustly enhanced the activities of key antioxidant enzymes (SOD, CAT, and GSH-Px) in the renal tissue while simultaneously reducing oxidative damage markers (MDA), thereby confirming its potent antioxidant capacity.

As a fundamental component of cellular defence mechanisms, CAT plays an indispensable role in stress adaptation and disease resistance [71]. Through coordinated action with GSH-Px, CAT efficiently eliminates H_2_O_2_ generated by SOD from •O^2−^ radicals, thereby protecting the cellular constituents from peroxide-mediated toxicity [72]. The concentration of lipid peroxidation end products (notably MDA) provides a direct quantitative measure of oxidative stress intensity [73], rendering MDA levels, together with SOD/CAT activity profiles, reliable biomarkers of ageing progression. T-AOC represents an integrative measure of systemic antioxidant defence and is particularly significant in ageing research [74]. The proinflammatory cytokine TNF-α features prominently in the pathogenesis of various chronic and autoimmune conditions [75]. Our experimental results demonstrate that *D. officinale* treatment significantly elevated levels of SOD, GSH-Px, CAT, and T-AOC in both the renal tissue and serum, while concurrently reducing concentrations of MDA and TNF-α, indicating comprehensive antioxidant and cytoprotective effects against D-Gal-induced damage.

Organ indices provided valuable parameters for assessing the validity of the ageing model (Figure 8). *D. officinale* treatment increased the organ-to-body weight ratio, indicating enhanced immune function in ageing models. Although D-Gal administration markedly suppressed body weight gain in the control animals, *D. officinale* intervention restored body mass to varying degrees, demonstrating its capacity to counteract multiple aspects of age-related physiological decline.

It should be noted that although the *D. officinale* extract used in this study was prepared using a previously reported standardised method [25], quantitative analysis of the chemical composition of the experimental batch was not conducted, which may affect the interpretation and reproducibility of the results. Future studies should employ techniques such as HPLC-DAD or UHPLC-MS/MS to quantitatively detect the major active components in the experimental batch of extract—including compounds predicted by network pharmacology, such as 5,7-dihydroxyflavone, mangiferic acid, and naringenin—in order to establish more robust quality control standards. Additionally, this study has several other limitations. First, while the D-galactose-induced aging model has been widely used for preliminary anti-aging efficacy evaluation owing to its practicality and reproducibility, it does not fully recapitulate the complex pathophysiology of natural aging processes. Second, the absence of a control group treated with the extract alone (without D-Galactose) limits our ability to distinguish between genuine restorative effects in the aged state and general physiological effects of the extract. While this study demonstrated the potential anti-ageing effects of *D. officinale* extracts in a D-Gal-induced mouse model, the findings are limited by the model’s incomplete representation of natural ageing processes and the lack of long-term efficacy and safety data. Furthermore, although we identified key targets and pathways via network pharmacology and molecular docking, these predictions require experimental validation (e.g., mechanistic studies) to confirm their biological relevance. Additionally, the reliance on ethanol extracts may have led us to overlook potential synergistic effects from alternative extraction methods. Furthermore, this study has limitations in the scope of inflammatory biomarkers assessed. Although TNF-α serves as a key mediator of systemic inflammation, it does not fully capture the complexity of inflammatory networks. Future studies should consider incorporating a broader panel of biomarkers—such as IL-6 (an acutely responsive pro-inflammatory cytokine), MCP-1 (a monocyte chemoattractant associated with immune cell infiltration), and ICAM-1 (an endothelial activation marker)—to provide a more comprehensive understanding of the extract’s anti-inflammatory mechanisms. Such multi-dimensional profiling would significantly enhance the depth of mechanistic interpretation and increase the translational relevance of our findings.

## 5. Conclusions

In summary, this study provides evidence that *D. officinale* may exert anti-ageing effects through interactions involving multiple bioactive components, molecular targets, and signalling pathways, particularly those related to oxidative stress and inflammatory responses. It should be noted that this study has certain limitations. For instance, a systematic analysis of the pharmacokinetic behaviour of the bioactive compounds in vitro and in vivo has not been conducted, the validation of cellular signalling pathways still relies primarily on indirect evidence, and the functional verification of component–target interactions under in vivo conditions requires further substantiation. Future studies should focus on clinical translation and more detailed mechanistic investigations to fully evaluate the therapeutic potential of this herb.

## Figures and Tables

**Figure 1 foods-14-03418-f001:**
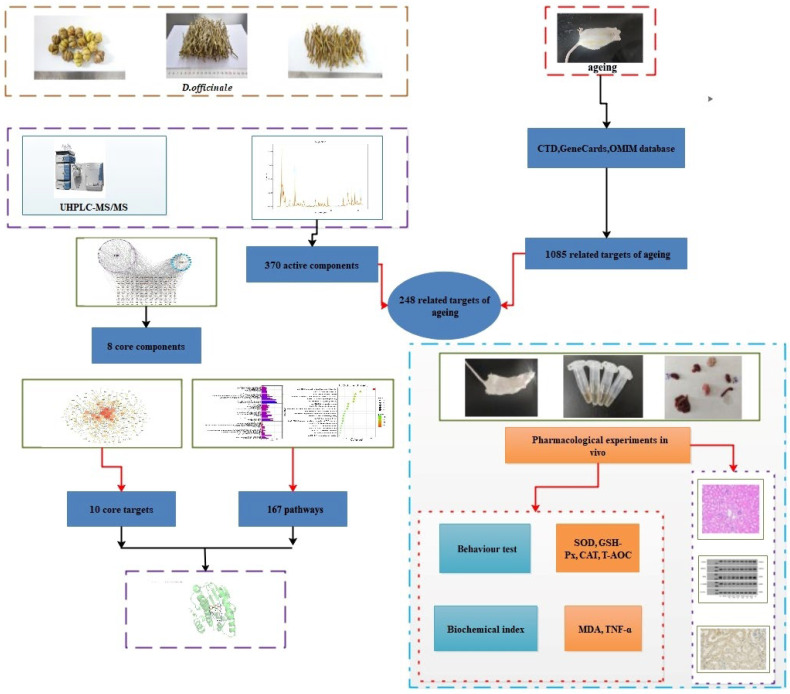
Flowchart of the study for investigating the anti-ageing effects of *Dendrobium officinale*.

**Figure 2 foods-14-03418-f002:**
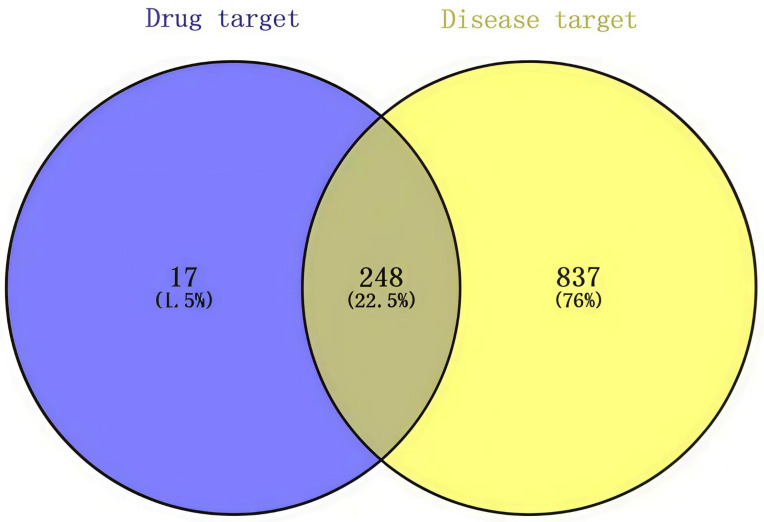
Venn diagram for prediction of ageing-related targets in Tiepi Fengdou therapy.

**Figure 3 foods-14-03418-f003:**
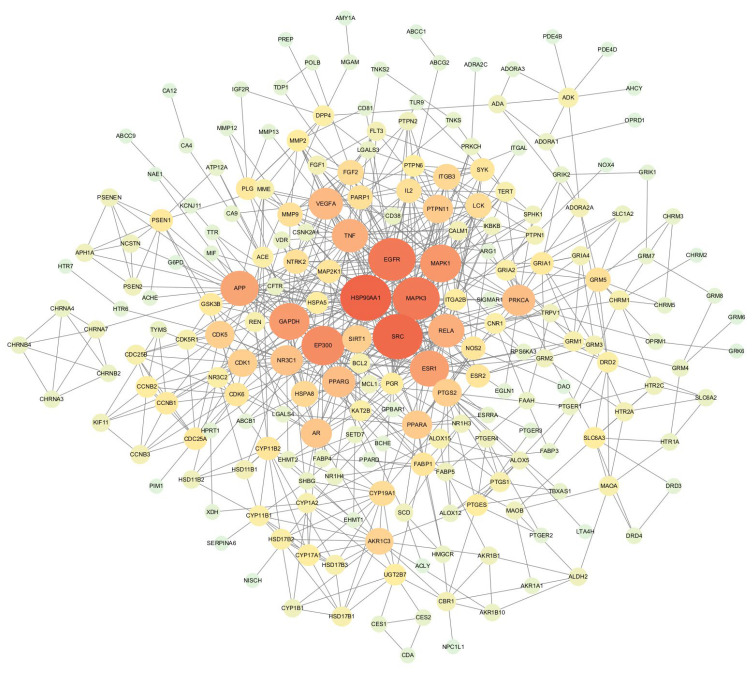
Protein–protein interaction (PPI) network analysis.

**Figure 4 foods-14-03418-f004:**
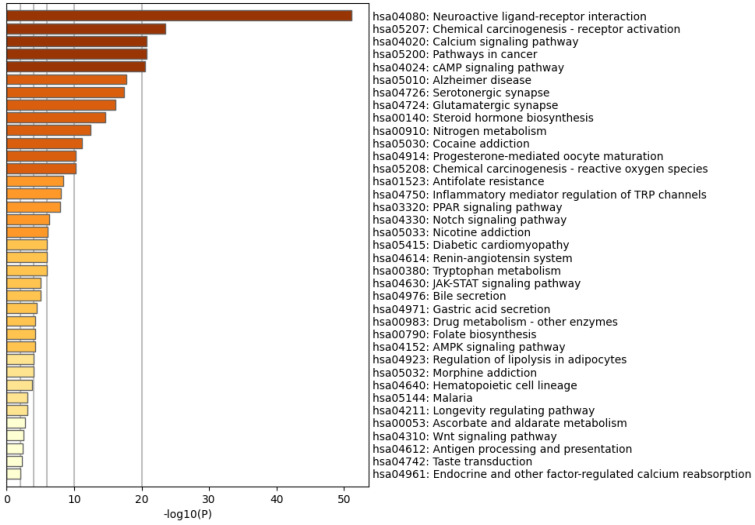
Gene ontology pathway enrichment analyses of 248 common target genes. (BP) Biological process; (CC) cellular component; (MF) molecular function.

**Figure 5 foods-14-03418-f005:**
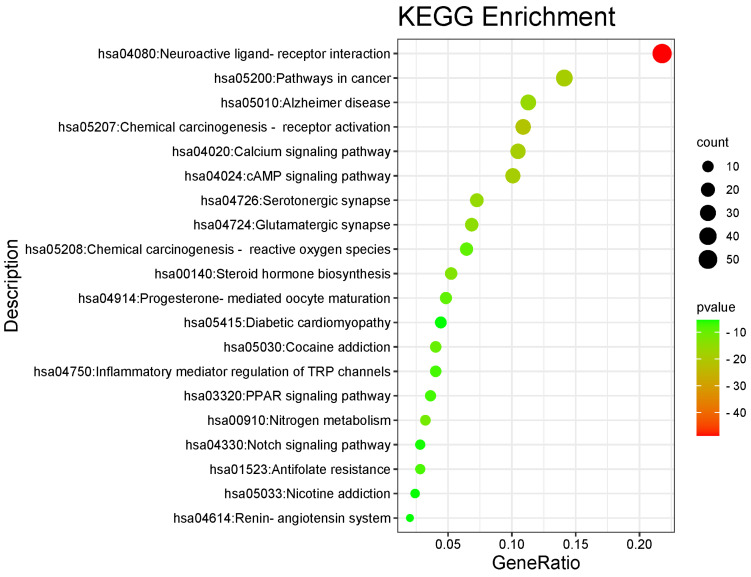
Top 20 pathways.

**Figure 6 foods-14-03418-f006:**
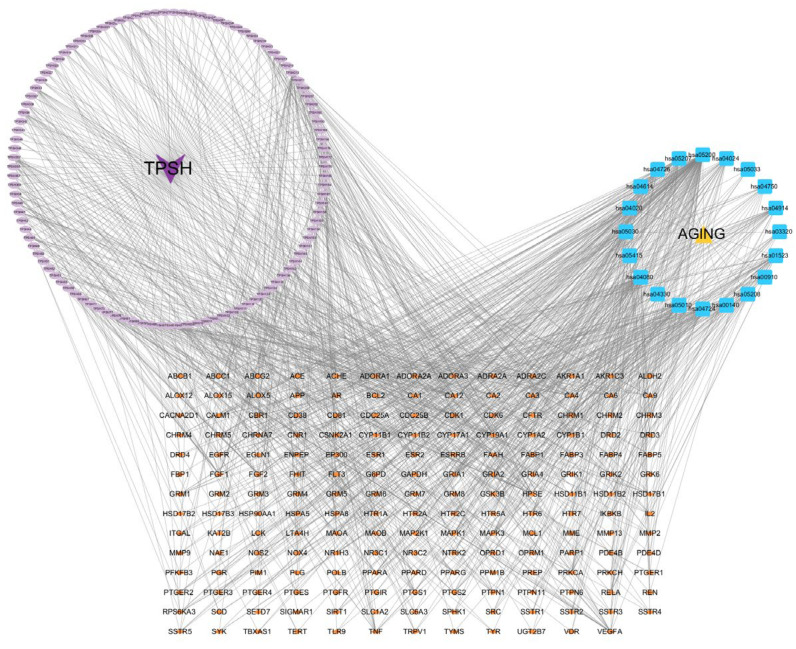
Component–target–pathway network of the *Dendrobium officinale* extract. Lilac oval nodes represent the component, orange small diamonds represent the targets, and blue nodes represent the pathways.

**Figure 7 foods-14-03418-f007:**
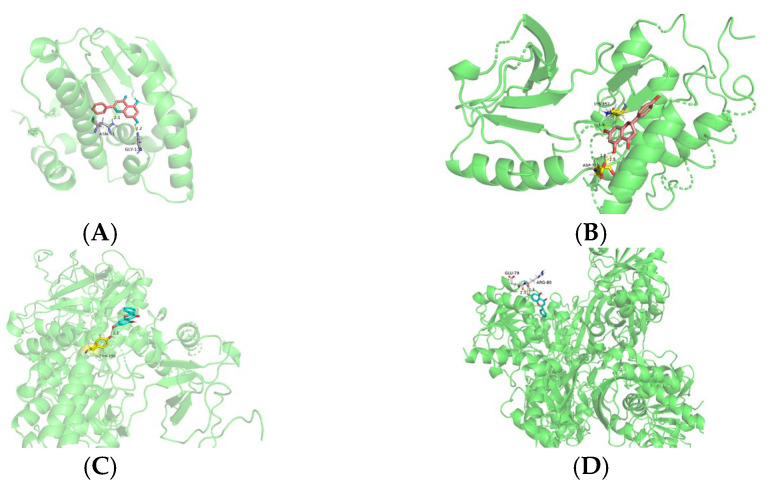
Schematic (3D) representation of the molecular model of the active component combined with target proteins. Component: 5,7-dihydroxyflavone. Target proteins: (**A**) HSP90AA1, (**B**) EGFR, (**C**) MAPK3, and (**D**) GAPDH. Components: norartocarpanone and naringenin. Target protein: HSP90AA1.

**Figure 8 foods-14-03418-f008:**
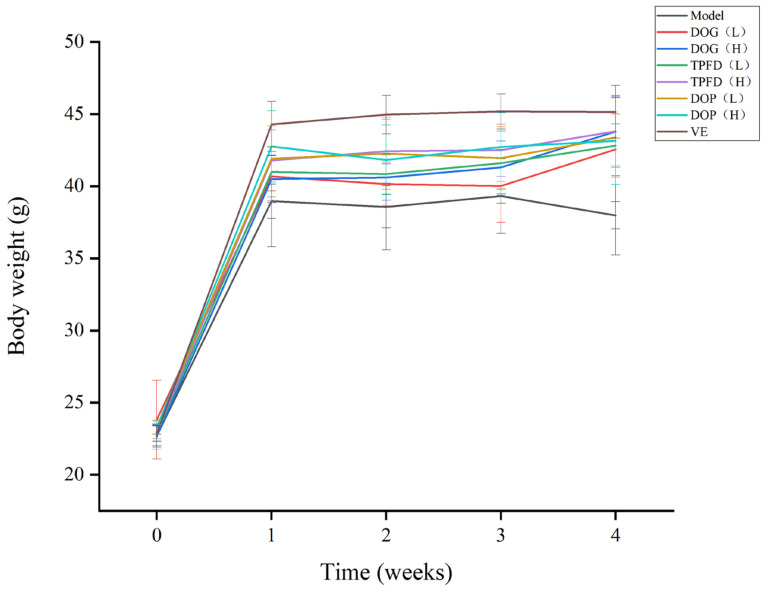
Effect of DOG, TPFD, and DOP on body weight.

**Figure 9 foods-14-03418-f009:**
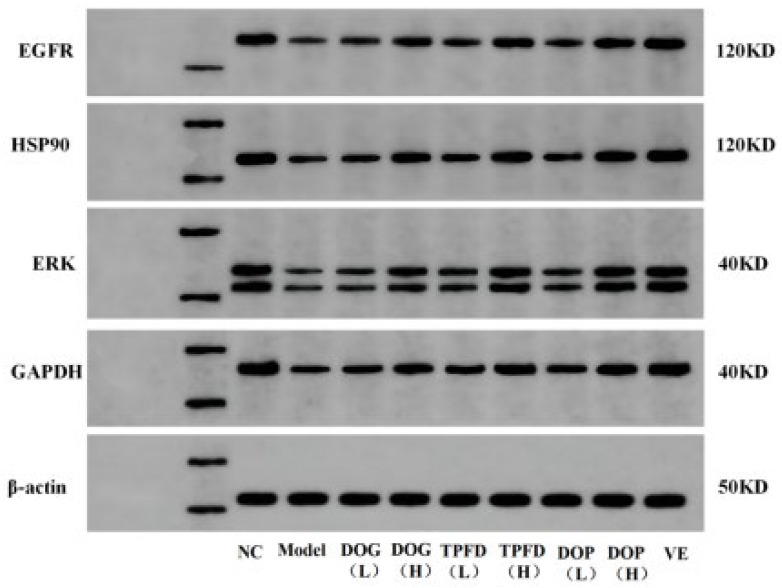
Western blotting analysis of the expression of EGFR, HSP90AA1, ERK, and GAPDH proteins.

**Figure 10 foods-14-03418-f010:**
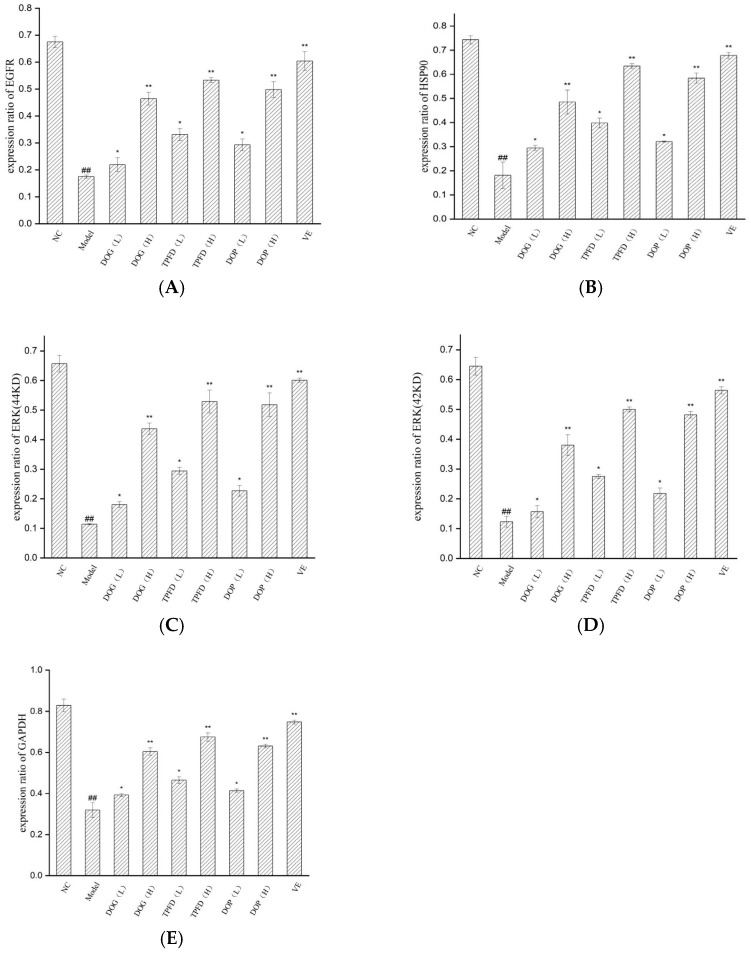
Results of Western blotting in kidney tissues. (**A**–**E**) The expression of EGFR, HSP90, ERK (44 and 42 KD), and GAPDH. The results are expressed as the means ± SD (*n* = 6). ## *p* < 0.01 compared with the control group. * *p* < 0.05, ** *p* < 0.01 compared with the model group.

**Figure 11 foods-14-03418-f011:**
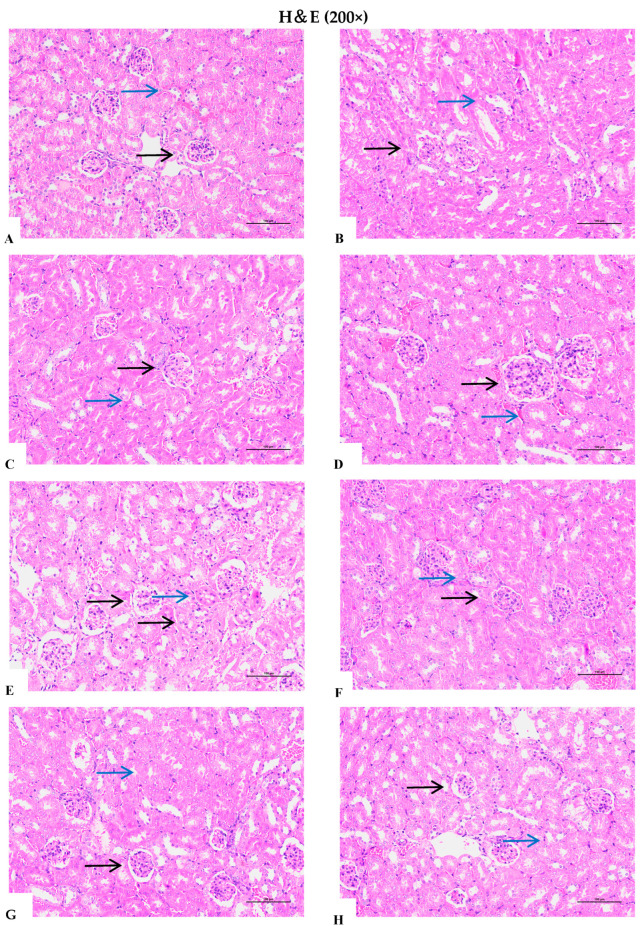
Effect of Dendrobium officinale on the ageing of kidneys in D-Galactose-induced ageing mice. Histopathological changes observed in D-Gal-induced mice. (**A**) Control group. (**B**) Model group. (**C**) DOG (L)-treated group. (**D**) DOG (H)-treated group. (**E**) TPFD (L)-treated group. (**F**) TPFD (H)-treated group. (**G**) DOP (L)-treated group. (**H**) DOP (H)-treated group. (**I**) VE-treated group. Black arrows indicate the glomerular morphology, highlighting the specific structural appearance and architecture for pathological assessment. Blue arrows point to the renal tubules. Histopathological changes were identified under a light microscope after haematoxylin and eosin staining (×200).

**Figure 12 foods-14-03418-f012:**
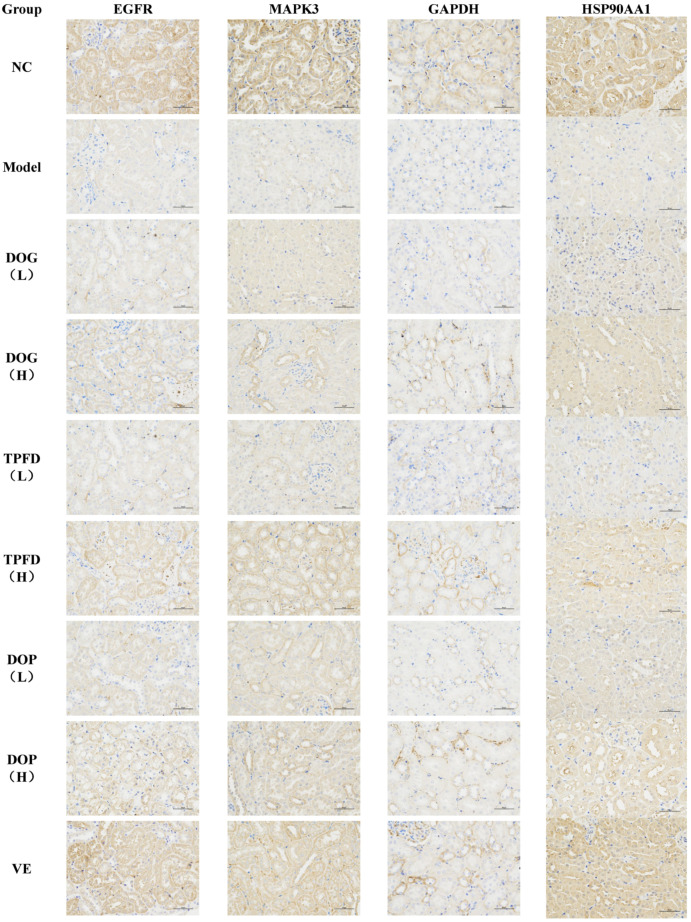
Immunohistochemical staining for EGFR, HSP90, ERK, and GAPDH proteins.

**Figure 13 foods-14-03418-f013:**
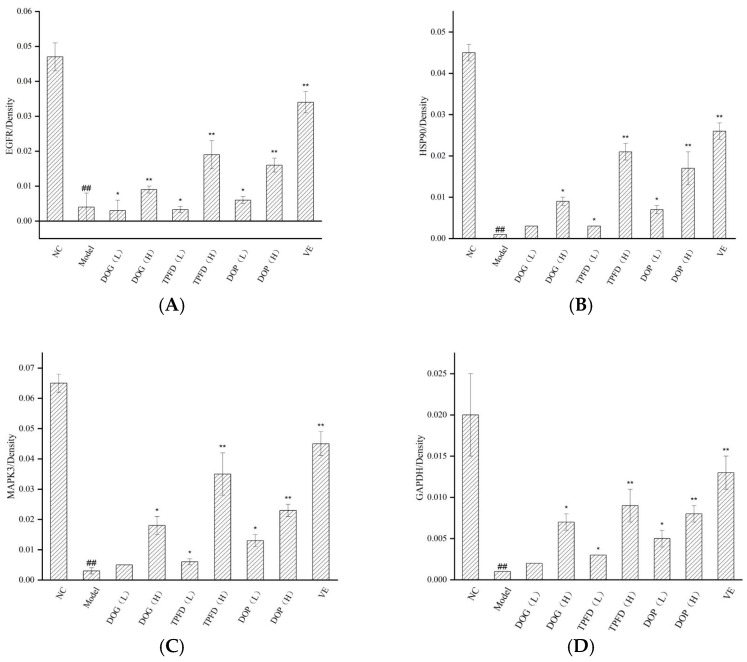
Expression of (**A**) EGFR, (**B**) HSP90, (**C**) ERK, and (**D**) GAPDH proteins based on densitometric analysis. ## *p* < 0.01 compared with the control group. * *p* < 0.05, ** *p* < 0.01 compared with the model group.

**Table 1 foods-14-03418-t001:** Organ indices of mice in each group ($\overset{-}{x}$ ± s, *n* = 10).

Experimental Group	Kidney Index (Left)	Kidney Index (Right)	Heart Index	Liver Index	Spleen Index	Lung Index	Brain Index
NC	0.78 ± 0.038	0.79 ± 0.05	0.54 ± 0.02	4.45 ± 0.18	0.31 ± 0.02	0.60 ± 0.02	1.18 ± 0.03
Model	0.50 ± 0.12 ^##^	0.52 ± 0.03 ^##^	0.38 ± 0.01 ^##^	3.21 ± 0.01 ^##^	0.19 ± 0.01 ^##^	0.44 ± 0.02 ^##^	0.82 ± 0.03 ^##^
DOG (L)	0.65 ± 0.028 **	0.64 ± 0.09 *	0.49 ± 0.02 **	3.72 ± 0.13	0.31 ± 0.04	0.55 ± 0.02 **	0.96 ± 0.04 **
DOG (H)	0.66 ± 0.026 **	0.68 ± 0.015 *	0.50 ± 0.04 **	3.88 ± 0.17 *	0.32 ± 0.04 *	0.59 ± 0.03 **	0.90 ± 0.02 **
TPFD (L)	0.62 ± 0.029 *	0.65 ± 0.02 *	0.48 ± 0.018 **	3.77 ± 0.12	0.28 ± 0.03	0.51 ± 0.02 *	0.88 ± 0.03
TPFD (H)	0.66 ± 0.025 **	0.67 ± 0.02 *	0.46 ± 0.04 **	3.63 ± 0.15	0.35 ± 0.07 *	0.60 ± 0.07 **	0.93 ± 0.03 **
DOP (L)	0.70 ± 0.04 **	0.71 ± 0.05 **	0.47 ± 0.02 **	4.03 ± 0.24 *	0.31 ± 0.05	0.53 ± 0.03 *	0.93 ± 0.03 **
DOP (H)	0.54 ± 0.02	0.61 ± 0.03 *	0.49 ± 0.03 **	3.84 ± 0.14 *	0.33 ± 0.08 *	0.53 ± 0.03 *	0.91 ± 0.04 **
VE	0.64 ± 0.02 **	0.67 ± 0.036 **	0.48 ± 0.02 **	4.48 ± 0.15 **	0.23 ± 0.02	0.56 ± 0.05 *	0.87 ± 0.02 *

Notes: ## *p* < 0.01 compared with the control group. * *p* < 0.05, ** *p* < 0.01 compared with the model group.

**Table 2 foods-14-03418-t002:** Serum total antioxidant capacity and TNF-α concentration ($\overset{-}{x}$ ± s, *n* = 10).

Group	T-AOC (mM)	TNF-α (ng/L)
NC	1.167 ± 0.069	133.576 ± 13.224
DOG (L)	0.312 ± 0.026 *	400.281 ± 10.392 *
Model	0.156 ± 0.055 ^##^	463.94 ± 19.443 ^##^
DOG (H)	0.722 ± 0.050 **	292.954 ± 10.291 **
TPFD (L)	0.548 ± 0.034 *	315.680 ± 21.696 *
TPFD (H)	0.940 ± 0.047 **	221.043 ± 19.267 **
DOP (L)	0.452 ± 0.041 *	359.179 ± 19.867 *
DOP (H)	0.849 ± 0.048 **	268.982 ± 7.093 **
VE	1.046 ± 0.054 **	173.581 ± 17.287 **

Notes: ## *p* < 0.01 compared with the control group. * *p* < 0.05, ** *p* < 0.01 compared with the model group.

**Table 3 foods-14-03418-t003:** Effect of *Dendrobium officinale* on the malondialdehyde (MDA) content, and glutathione peroxidase (GSH-Px), catalase (CAT), and superoxide dismutase (SOD) activities in the kidney tissue of mice ($\overset{-}{x}$ ± s, *n* = 10).

Group	MDA (nmol/mL)	GSH-Px (U/gprot)	CAT (U/mgprot)	SOD (U/mgprot)
NC	0.772 ± 0.192	30.465 ± 1.952	20.256 ± 1.190	37.943 ± 1.461
Model	3.987 ± 0.225 ^##^	7.317 ± 1.276 ^##^	3.913 ± 0.616 ^##^	10.054 ± 1.610 ^##^
DOG (L)	3.427 ± 0.229 *	11.353 ± 1.046 *	6.403 ± 0.880 *	14.262 ± 1.258 *
DOG (H)	2.215 ± 0.117 **	18.055 ± 1.290 **	12.091 ± 0.704 **	22.040 ± 1.429 **
TPFD (L)	2.864 ± 0.093 *	15.607 ± 1.131 *	9.875 ± 0.792 **	20.929 ± 1.890 *
TPFD (H)	1.526 ± 0.117 **	24.138 ± 1.730 **	15.795 ± 0.774 **	30.685 ± 0.904 **
DOP (L)	3.263 ± 0.149 *	15.988 ± 0.706 *	9.440 ± 0.719 *	16.518 ± 1.308 *
DOP (H)	1.803 ± 0.125 **	21.625 ± 0.565 **	13.638 ± 0.660 **	23.333 ± 1.593 **
VE	1.298 ± 0.094 **	27.054 ± 1.141 **	16.717 ± 1.031 **	32.935 ± 1.353 **

Notes: ## *p* < 0.01 compared with the control group. * *p* < 0.05, ** *p* < 0.01 compared with the model group.

## Data Availability

The original contributions presented in the study are included in the article/Appendix A. Further inquiries can be directed to the corresponding author(s).

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
