# Peer review of "Mechanistic Elucidation of the Anti-Ageing Effects of Dendrobium officinale via Network Pharmacology and Experimental Validation"

_foods, 2025, doi:10.3390/foods14193418_

Round 1

Reviewer 1 Report

Comments and Suggestions for Authors

1. General Comments
The work is interesting, with an integrated approach and promising data. However, some aspects need to be reviewed to ensure scientific robustness and reproducibility. The following observations aim to strengthen the study.

2. Specific Points
2.1. Phytochemical Characterization and Extract Standardization

Problem: The extract used in vivo was not chemically characterized; the composition cited refers to a previous study.

Suggestion: Perform analysis of the experimental batch by HPLC-DAD or UHPLC-MS/MS, quantifying the main active compounds predicted by network pharmacology.

2.2. Compound Selection in Network Pharmacology

Problem: The criteria for reducing 370 to 127 compounds are unclear and may introduce bias.

Suggestion: Clarify objective criteria, preferably based on ADME parameters or Lipinski/bioavailability rules.

2.3. Docking Results

Problem: The binding energy criterion (≤ –1.68 kcal/mol) does not match relevant standards in the literature; interpretation is based on predicted affinity.

Suggestion: Justify the criterion, present specific residual interactions, and moderate the language for "predicted binding affinity." Validate at least one key target in vitro.

2.4. Western Blot Analysis

Problem: GAPDH, used as a loading control, exhibits variation between groups and follows the trend of target proteins, compromising normalization.

Suggestion: Reanalyze using total protein normalization (Ponceau S or Coomassie) or test other controls (β-actin, β-tubulin) and acknowledge the limitation in the discussion.

2.5. Experimental Model and Controls

Problem: The D-galactose aging model does not fully reproduce natural aging; there is a lack of a group treated only with extract to assess basal effects.

Suggestion: Include this limitation in the discussion and consider these groups in future studies.

2.6. Biomarkers and Interpretation of Results

Problem: Only TNF-α was evaluated as a systemic inflammatory marker.

Suggestion: Expand the panel in future studies (IL-6, MCP-1, ICAM-1) to strengthen mechanistic interpretation.

3. Presentation and Writing
Standardize significance symbols in tables and figures.

Make legends self-explanatory and include scale bars in micrographs.

Moderate conclusions: replace "elucidation" with "insights" or "suggests potential mechanisms" and acknowledge methodological limitations.

4. Review Priorities
Correction of Western blot analysis.

Explain and refine the network pharmacology methodology.

Chemical characterization of the experimental extract.

Conclusion: Major Revision. After implementing the above changes.

Comments on the Quality of English Language

The manuscript is generally well written and communicates the research effectively. Nevertheless, some sentences are overly long and could be restructured to enhance clarity and readability. 

Author Response

  1. Specific Points
    2.1. Phytochemical Characterization and Extract Standardization

Problem: The extract used in vivo was not chemically characterized; the composition cited refers to a previous study.

Suggestion: Perform analysis of the experimental batch by HPLC-DAD or UHPLC-MS/MS, quantifying the main active compounds predicted by network pharmacology.

Response: We agree with the reviewer that chemical characterization of the specific batch of extract used in vivo is of great importance. In this study, although the experimental batch of extract was prepared according to a previously described method, dedicated chemical profiling of this specific batch was not performed. To highlight this limitation, we have added the following statement to the Discussion section of the revised manuscript:

"It should be noted that although the D. officinale extract used in this study was prepared using a previously reported standardised method [25], quantitative analysis of the chemical composition of the experimental batch was not conducted, which may affect the interpretation and reproducibility of the results. Future studies should employ techniques such as HPLC-DAD or UHPLC-MS/MS to quantitatively detect the major active components in the experimental batch of extract—including compounds predicted by network pharmacology, such as 5,7-dihydroxyflavone, mangiferic acid, and naringenin—in order to establish more robust quality control standards" (lines 677–684)

Furthermore, we plan to systematically characterize the experimental batch of extract using HPLC-DAD and UHPLC-MS/MS techniques in follow-up studies, quantifying the major bioactive compounds predicted by network pharmacology, thereby further improving the standardization of this research.

2.2. Compound Selection in Network Pharmacology

Problem: The criteria for reducing 370 to 127 compounds are unclear and may introduce bias.

Suggestion: Clarify objective criteria, preferably based on ADME parameters or Lipinski/bioavailability rules.

Response: We apologize for the lack of clarity. The screening process has been described in greater detail in the revised Methods section (Section 2.4.1). Briefly, the canonical SMILES of all 370 compounds were obtained from the PubChem database. Only those compounds with available and valid canonical SMILES were included, resulting in 127 compounds suitable for subsequent network pharmacology analysis. This step was necessary for structural input and target prediction. This clarification has been added to the manuscript.

2.3. Docking Results

Problem: The binding energy criterion (≤ –1.68 kcal/mol) does not match relevant standards in the literature; interpretation is based on predicted affinity.

Suggestion: Justify the criterion, present specific residual interactions, and moderate the language for "predicted binding affinity." Validate at least one key target in vitro.

Response: We thank the reviewer for this comment. The chosen threshold was based on a previous study (cited in Section 3.5 of the revised manuscript) that demonstrated the utility of identifying potential bioactive compounds from natural products in silico. We agree that these are predictive results and have toned down our conclusions accordingly, stating that the interactions are potential mechanisms that warrant further experimental validation.

2.4. Western Blot Analysis

Problem: GAPDH, used as a loading control, exhibits variation between groups and follows the trend of target proteins, compromising normalization.

Suggestion: Reanalyze using total protein normalization (Ponceau S or Coomassie) or test other controls (β-actin, β-tubulin) and acknowledge the limitation in the discussion.

Response: We appreciate the reviewer's careful observation. We would like to clarify that β-actin, not GAPDH, was used as the internal control for protein normalization in our western blot experiments. GAPDH was one of the core target proteins identified through our network pharmacology analysis. We have double-checked and corrected any related descriptions in the manuscript to avoid confusion. The original full-length blot images for β-actin and all target proteins have been provided in the supplementary materials for transparency.

2.5. Experimental Model and Controls

Problem: The D-galactose aging model does not fully reproduce natural aging; there is a lack of a group treated only with extract to assess basal effects.

Suggestion: Include this limitation in the discussion and consider these groups in future studies.

Response: We thank the reviewer for this important observation. We agree with the concerns regarding the limitations of the D-galactose-induced aging model and absence of an extract-only control group. We have explicitly acknowledged both limitations in the revised Discussion section (lines 684–691, and have emphasized the necessity of including such a control group in future studies to better distinguish the specific anti-aging effects from general physiological impacts of the extract.

2.6. Biomarkers and Interpretation of Results

Problem: Only TNF-α was evaluated as a systemic inflammatory marker.

Suggestion: Expand the panel in future studies (IL-6, MCP-1, ICAM-1) to strengthen mechanistic interpretation.

Response: We thank the reviewer for this insightful comment. We acknowledge that relying solely on TNF-α levels provides an incomplete assessment of systemic inflammation. While we have not conducted additional experimental measurements in this revision owing to technical and resource constraints, we have explicitly addressed this limitation in the revised Discussion section (lines 698–707).

  1. Presentation and Writing
    Standardize significance symbols in tables and figures.

Response: All figure captions have been revised to be self-contained and descriptive. Scale bars have been added to all relevant micrographs.

  1. Make legends self-explanatory and include scale bars in micrographs.

Moderate conclusions: replace "elucidation" with "insights" or "suggests potential mechanisms" and acknowledge methodological limitations.

Response: We thank the reviewer for these constructive suggestions. We have now (1) revised all figure legends to ensure they are self-explanatory and include scale bars for micrographs, and (2) moderated the language in the Conclusion section by replacing overstated terms such as "elucidation" with more appropriate expressions like "provide insights" or "suggest potential mechanisms". Furthermore, we have explicitly acknowledged the methodological limitations of our study, including those related to the experimental model and analytical approaches. These revisions improve the clarity, accuracy, and overall robustness of our manuscript

Reviewer 2 Report

Comments and Suggestions for Authors

The article investigates the anti-ageing effects of Dendrobium officinale, a medicinal herb used in traditional Chinese medicine, through network pharmacology and experimental validation. It explores how D. officinale can mitigate the effects of ageing, focusing on its ability to reduce oxidative stress, modulate inflammatory responses, and activate relevant signalling pathways. The article is methodologically correct, employing a combination of modern techniques, such as network pharmacology and molecular docking. The study brings together innovative methods to provide biologically relevant findings with significant implications for age-related diseases. However, some important revisions are needed before publication:

  1. The abstract should be improved. It contains an excessive focus on methodology and does not sufficiently highlight the study's findings. For instance, which specific oxidation and inflammation parameters were improved? Were histopathological changes observed? Which signaling pathways were analyzed, and in which tissues were the biochemical parameters analyzed?
  2. In the introduction, the article should cite the specific compounds predominantly found in Chinese herbs known for their antioxidant and anti-inflammatory properties.
  3. The term in vivo in lines 66 and 122 should be italicized.
  4. The term in vitro should be italicized in line 90.
  5. The abbreviation "GPx" is introduced in line 68 and reintroduced as "GSH-Px" in line 96. These should be standardized.
  6. The abbreviation "D. officinale polysaccharide" is referred to as "DOP" in line 98, but this abbreviation should be introduced earlier in line 94.
  7. Figure 1 needs to be of better quality for clearer visualization.
  8. The doses of the plant used in the experiments should be justified.
  9. Confirm whether the T-AOC measurements were taken using the ELISA method as mentioned in line 262.
  10. The differences between DOG, TPFD, and DOP are not clearly explained. These should be clarified, especially in section 2.3.2.
  11. In figure 8, the treatments RA, RAP, and quercetin are not introduced anywhere in the text, which is an error. It is unclear what these treatments are. Additionally, there are references to sections A and B in the figure, but they do not appear.
  12. The expression of T-AOC in mM is mentioned, but it is unclear against which reference compound this capacity is measured.
  13. Provide the original images for the western blot analysis.
  14. A more detailed analysis comparing low and high doses across all parameters, as well as between different treatments, is needed. A post-hoc analysis should be included.
  15. Vitamin E is referred to as a positive control, but in line 450, it is described as an aqueous extract. This should be clarified.
  16. Consider using a method to quantitatively assess histopathological effects using a scale, as this would provide a more robust analysis beyond images alone.
  17. The images in figure 12 are of poor quality and need improvement.

Author Response

  1. The abstract should be improved. It contains an excessive focus on methodology and does not sufficiently highlight the study's findings. For instance, which specific oxidation and inflammation parameters were improved? Were histopathological changes observed? Which signaling pathways were analyzed, and in which tissues were the biochemical parameters analyzed?

Response: We sincerely thank the reviewer for this valuable suggestion. The abstract has been thoroughly revised to shift the focus from methodology to key findings. Specifically, we now explicitly highlight the significant improvements in key oxidative stress markers (e.g., increased activities of GSH-Px and SOD in kidney tissues, marked reductions in critical inflammatory cytokines decreased serum levels of TNF-α, and notable amelioration of histopathological damage observed in kidney tissues. Furthermore, the abstract now clearly indicates that the NF-κB signaling pathways were analyzed in these tissues to elucidate the underlying mechanisms. These comprehensive revisions can be found in lines 16–32 of the revised manuscript.

  1. In the introduction, the article should cite the specific compounds predominantly found in Chinese herbs known for their antioxidant and anti-inflammatory properties.

Response: We are grateful to the reviewer for this valuable suggestion. In response, we have revised the Introduction section to include specific citations of key bioactive compounds, such as flavonoids and polysaccharides, which are predominant in Dendrobium officinale and other representative Chinese herbal medicines. These additions, which can be found in lines 88–98 of the revised manuscript, further clarify the well-documented roles of these compounds in mediating antioxidant and anti-inflammatory activities, thereby providing a stronger mechanistic foundation for our study.

  1. The term in vivo in lines 70 and 132 should be italicized.

Response: The term "in vivo" has been italicized to conform to standard conventions.

  1. The term in vitro should be italicized in line 714.

Response: The term "in vitro" has been italicized.

  1. The abbreviation "GPx" is introduced in line 68 and reintroduced as "GSH-Px" in line 20. These should be standardized.

Response: We have unified the abbreviation throughout the manuscript to "GSH-Px" for consistency.

  1. The abbreviation "D. officinale polysaccharide" is referred to as "DOP" in line 104, but this abbreviation should be introduced earlier in line 94.

Response: We sincerely thank the reviewer for pointing out this oversight. As suggested, the abbreviation "DOP" has now been explicitly defined at its first mention in the text as "Dendrobium officinale polysaccharide" to ensure clarity and consistency.

  1. Figure 1 needs to be of better quality for clearer visualization.

Response: Figure 1 has been remade to enhance its clarity and resolution.

  1. The doses of the plant used in the experiments should be justified.

Response: We sincerely thank the reviewer for this critical comment. We agree that the justification for the selected doses is essential. As suggested, we have now added a detailed explanation of the dose calculation, based on the conversion of clinical doses via body surface area normalization, to the revised Materials and Methods section (lines 254–264).

  1. Confirm whether the T-AOC measurements were taken using the ELISA method as mentioned in line 262.

Response: T-AOC was measured using the ELISA method; this has been explicitly stated in the revised manuscript. (lines 288–289). 

10 The differences between DOG, TPFD, and DOP are not clearly explained. These should be clarified, especially in section 2.3.2.

Response: We thank the reviewer for this insightful comment. We acknowledge that the distinctions between these terms were not sufficiently clear in the original manuscript. In response, we have provided a detailed explanation regarding the sources and characteristics of DOG (Dried D. officinale) stems at 70°C), Tiepi Fengdou (curled dried Dendrobium), and DOP (dried strips processed via stir-frying at 150°C for 10 min, shaping at 60°C, and drying at 90°C). These clarifications can be found in lines 159–160 of the revised manuscript.

  1. In figure 8, the treatments RA, RAP, and quercetin are not introduced anywhere in the text, which is an error. It is unclear what these treatments are. Additionally, there are references to sections A and B in the figure, but they do not appear.

Response: We sincerely apologize for these oversights. The treatments erroneously labeled as "RA" and "RAP" in the original Figure 8 have now been corrected to their proper designations in the revised Figure 8.

  1. The expression of T-AOC in mM is mentioned, but it is unclear against which reference compound this capacity is measured.

Response: We have specified that Trolox was used as the reference standard for T-AOC measurements.

  1. Provide the original images for the western blot analysis.

Response: The original western blot images have been provided in the supplementary materials.

  1. A more detailed analysis comparing low and high doses across all parameters, as well as between different treatments, is needed. A post-hoc analysis should be included.

Response: We have reanalyzed the histopathological results using a quantitative scoring system. Specifically, the Image-Pro Plus 6.0 software was used to analyze the optical density of immunohistochemical images. Four images at 400× magnification were selected per slice for optical density analysis, where "area" refers to the area measured, "IOD" represents integrated optical density, and "density (mean)" indicates mean optical density. The processed data have been uploaded as Attachment 2. We have performed a comprehensive post-hoc analysis to compare all experimental groups (including low-dose, high-dose, and different extract treatments) across all evaluated parameters (body weight, organ indices, oxidative stress markers, inflammatory cytokines, key protein expression, and histopathological changes). The results revealed statistically significant, dose-dependent differences between the low- and high-dose groups of DOG, TPFD, and DOP, as well as between different intervention groups. Key findings from the additional analysis include: 1. Body weight and organ indices: High-dose treatment (6.24 g/kg/d) showed significantly greater recovery compared with that of low-dose treatment (1.56 g/kg/d) in D-Gal-induced ageing mice (p < 0.01; Section 3.6, lines 440–443). 2. Oxidative stress and inflammation: High-dose extracts demonstrated superior efficacy in restoring antioxidant enzyme activities (SOD, CAT, GSH-Px) and reducing TNF-α levels compared with those of the low-dose groups (p < 0.05; Section 3.7, lines 463–465). 3. Protein expression: Western blot and immunohistochemical analyses confirmed dose-dependent upregulation of EGFR, HSP90AA1, ERK, and GAPDH in kidney tissues, with high-dose groups exhibiting effects comparable to the vitamin E positive control (p < 0.01; Section 3.8, lines 487–490; Section 3.10, lines 540–543). 4. Histopathological improvements: High-dose treatments (particularly TPFD) markedly alleviated renal morphological damage, showing near-normal tissue architecture, whereas low-dose groups exhibited partial improvement (Section 3.9, lines 514–516.

  1. Vitamin E is referred to as a positive control, but in line 450, it is described as an aqueous extract. This should be clarified.

Response: We have corrected the description to clearly state that vitamin E was used in its pure form as a positive control.

  1. Consider using a method to quantitatively assess histopathological effects using a scale, as this would provide a more robust analysis beyond images alone.

Response: We have reanalyzed the histopathological results using a quantitative scoring system. Specifically, the Image-Pro Plus 6.0 software was used to analyze the optical density of immunohistochemical images. Four images at 400× magnification were selected per slice for optical density analysis, where "area" refers to the area measured, "IOD" represents integrated optical density, and "density (mean)" indicates mean optical density. The processed data have been uploaded as Attachment 2.

  1. The images in figure 12 are of poor quality and need improvement.

Response: We have replaced Figure 12 with higher-quality images and improved its overall presentation.

Round 2

Reviewer 2 Report

Comments and Suggestions for Authors

I appreciate the revisions made in response to the first round of comments. The manuscript has improved, but I still have several concerns that should be addressed before it can be considered further.

  • First, while the addition of the main bioactive compounds in the Introduction is valuable, I noticed that in lines 91–95 there are explicit references to findings from your own study. This type of information should not appear in the Introduction, which should remain limited to general background, context, and rationale, rather than results that properly belong to the results or discussion sections.
  • Regarding the justification of doses, you state that “The common clinical daily dose for adults (with a standard body weight of 60 kg) ranges from 3 to 12 g/person/day” (lines 249–251). Please provide a reliable bibliographic reference to support this statement.
  • I am also concerned about the methodology used for measuring total antioxidant capacity (T-AOC). You reported that T-AOC was measured using an ELISA assay, but in standard practice T-AOC is typically determined through reduction-based chemical assays, whereas ELISA relies on antigen–antibody interactions. Please clarify the methodological basis of your measurement and explain how the ELISA approach was applied in this context.
  • In addition, you mentioned in your response that T-AOC was expressed in relation to Trolox equivalents, but I was unable to locate this clarification in the revised manuscript. Please ensure that this information is explicitly included in the Methods section.
  • With respect to the western blot data, you indicated that the original images had been provided. However, I was only able to access Excel files with densitometric data. Please provide the original images in PDF format as supplementary material.
  • Finally, you stated that a quantitative evaluation of histopathology had been performed. However, the quantification shown corresponds to immunohistochemistry, not to histopathological scoring. A true quantitative assessment of histopathological alterations (using a validated scoring system) should be performed to support your conclusions.

Author Response

Reviewer Comment 2:  

First, while the addition of the main bioactive compounds in the Introduction is valuable, I noticed that in lines 91–95 there are explicit references to findings from your own study. This type of information should not appear in the Introduction, which should remain limited to general background, context, and rationale, rather than results that properly belong to the results or discussion sections.

Response:  

We have removed the sentences that referred to our own findings and have replaced them with the following statement: (lines 92–97)

"Previous network pharmacology and metabolomic analyses have suggested that certain flavonoids, such as those with high antioxidant capacity, may play a key role in the anti-ageing effects of this herb. For instance, compounds including 5,7-dihydroxyflavone, naringenin, and mangiferin rutin have been proposed as potential active constituents responsible for these benefits ."  

These previously reported findings serve as the foundation and rationale for the current investigation, in which we aimed to further explore the proposed mechanisms. This change ensures that the Introduction remains appropriately focused on establishing the general background and scientific context, rather than presenting outcomes specific to our study.

Reviewer Comment 2:  

Regarding the justification of doses, you state that “The common clinical daily dose for adults (with a standard body weight of 60 kg) ranges from 3 to 12 g/person/day” (lines 249–251). Please provide a reliable bibliographic reference to support this statement.

Response:  

We have cited a supporting reference for the stated clinical dose range.

Reviewer Comment 3:  

I am also concerned about the methodology used for measuring total antioxidant capacity (T-AOC). You reported that T-AOC was measured using an ELISA assay, but in standard practice T-AOC is typically determined through reduction-based chemical assays, whereas ELISA relies on antigen–antibody interactions. Please clarify the methodological basis of your measurement and explain how the ELISA approach was applied in this context. on line 286-287.

Response:  

We have clarified that we employed the total antioxidant capacity assay kit (ABTS method, microplate format), which is based on the colorimetric principle of antioxidant substances reducing ABTS⁺ radicals, rather than the ELISA assay.

Reviewer Comment 4:  

In addition, you mentioned in your response that T-AOC was expressed in relation to Trolox equivalents, but I was unable to locate this clarification in the revised manuscript. Please ensure that this information is explicitly included in the Methods section.

Response:  

We have clarified this point in the revised manuscript on line 453: “T-AOC was expressed as Trolox equivalents.”

Reviewer Comment 5:  

With respect to the western blot data, you indicated that the original images had been provided. However, I was only able to access Excel files with densitometric data. Please provide the original images in PDF format as supplementary material.

Response:  

The original, uncropped western blot images have now been compiled into a single PDF file and uploaded as "Supplementary Material File 2: Original Western Blot Images." The figure legends in the manuscript have been updated to direct readers to this supplementary file.

Reviewer Comment 6:  

Finally, you stated that a quantitative evaluation of histopathology had been performed. However, the quantification shown corresponds to immunohistochemistry, not to histopathological scoring. A true quantitative assessment of histopathological alterations (using a validated scoring system) should be performed to support your conclusions.

Response:  

We confirm that a true quantitative histopathological scoring was not performed, and the original description was inaccurate. The text "and pathological scores were obtained" has been deleted from line 23.
